# Practices and preferences for HIV testing and treatment services amongst partners of transgender women in Lima, Peru: An exploratory, mixed methods study

Claudia Kazmirak[1,2], Deanna Tollefson[2]*, Alexander Lankowski[2,3], Hugo Sanchez[4], Ivan Gonzales[4], Dianne Espinoza[5], Ann Duerr[2,3]

1 University of Illinois College of Medicine, Chicago, IL, United States of America, 2 Vaccine Infectious Disease Division, Fred Hutchinson Cancer Center, Seattle, WA, United States of America, 3 Department of Global Health, University of Washington, Seattle, WA, United States of America, 4 Epicentro Salud, Lima, Perú, 5 Universidad Peruana Cayetano Heredia, Lima, Perú

☯ These authors contributed equally to this work.
* dtollefs@fredhutch.org

**Data Availability Statement:** Data are included in the submission.

## Abstract

### Background

In Peru, one-third of transgender women (TW) are estimated to be living with HIV. While TW are recognized as a priority population, their sexual partners are an at-risk hidden population with unmet needs for HIV services. We conducted a study examining the practices and preferences for HIV services among partners of transgender women (PTW), as compared to TW, to better understand the needs of PTW and inform HIV service delivery for them in Peru.

### Methods

Between July-October 2022 we conducted a cross-sectional mixed methods study among PTW and TW in Lima, Peru. Using an explanatory sequential design, we administered online surveys to PTW (n = 165) and TW (n = 69), then interviewed a subset of participants (n = 20: 16 PTW, 4 TW). We quantitatively and qualitatively described PTW practices/perspectives on HIV testing and treatment and compared them to TW practices/preferences; we also compared practices/preferences among PTW based on their relationship with TW.

### Results

Overall, PTW and TW shared similar experiences and preferences for HIV testing/treatment, but fewer PTW reported accessing non-traditional HIV testing options and PTW expressed less strong preferences for HIV services. PTW practices/preferences varied by type of relationship with TWs. Surveys and interviews highlighted a need to prioritize efficiency for HIV testing, eliminate gender/sexuality-based discrimination in healthcare settings, increase privacy when delivering HIV services, and increase awareness of pre-exposure prophylaxis.

**Funding:** Research reported in this publication was supported by an award to CK from the Fogarty International Center of the National Institutes of Health under grants 5D43TW009345-10 and 2D43TW009345-11 awarded to the Northern Pacific Global Health Fellows Program. We acknowledge support related to the use of ITHS REDCap to carry out the survey used in this study (UL1 TR002319, KL2 TR002317, and TL1 TR002318 from NCATS/NIH). There was no additional external funding received for this study. The funders were not involved in the design or conduct of the study; the content is solely the responsibility of the authors and does not represent the views of the National Institutes of Health.

**Competing interests:** The authors have declared no competing interests.

## Conclusion

PTW identified many aspects related to the location, convenience, and privacy of HIV services as important. Next steps could include a discrete choice experiment to further clarify priorities for HIV services for PTW in Peru.

## Introduction

Transgender women (TW) are at increased risk of HIV infection globally [1, 2], which means their sexual partners are also at higher risk for acquiring HIV than the general population [3]. While recent years have seen an increase in efforts to improve HIV prevention and care delivery for TW, the need to prioritize their sexual partners in HIV programs has only recently gained attention [1, 3–6]. Partners of transgender women (PTW) are a hidden population who frequently identify as heterosexual cisgender males and often do not identify with the LGBTQ + community [3, 7–12]. Additionally, many PTW may not be publicly open about having TW partners due to stigma surrounding these relationships [3, 13]. As such, PTW tend to be missed by HIV prevention programs in countries with concentrated HIV epidemics, like Peru, which tend to target the LGBTQ+ community [3, 8, 9, 11, 14].

In Peru, one-third of TW are estimated to be living with HIV, resulting in high levels of HIV transmission within their sexual networks (i.e., with PTW) [15, 16]. Extensive research by Long et al. has demonstrated that PTW in Peru are at high risk for HIV, and that research is required to understand how to best reach them with HIV prevention and treatment services, as they are not part of the MSM community [7–10, 14]. The invisibility and heterogeneity of PTW has made it difficult to assess their knowledge of HIV, self-perceived HIV risk, and history of, or preferences for, HIV testing services [7, 8, 10, 12]. This is problematic, as such information is key to developing effective programs tailored to the needs of this historically underserved population. Traditional HIV testing, prevention, and treatment services in Peru are offered to anyone through the national health system at CERITS (government-run STI clinics) and there has been substantial outreach to reach TW through non-traditional services (community-based or venue-based testing) [17, 18]. However, neither traditional nor non-traditional HIV services have specifically targeted PTW, thus past research must be expanded to understand how to best reach this population with HIV prevention and care services [7–10, 14]. In addition, the heterogeneity of PTW in Peru, as highlighted by past research [7, 8], highlights the additional need to explore how such services should be tailored to reach PTW based on the type of relationship they have with TW. The need to understand how to best reach PTW in Peru is especially urgent, as the inclusion of PrEP to the country's national HIV prevention guidelines in June 2023 ushers in opportunities to offer HIV prevention services to both TW and PTW [19].

We therefore conducted this study to describe the practices and preferences that PTW in Peru have for HIV testing and treatment services and to determine how such practices and preferences (1) compare to those of TW and (2) compare amongst PTW based on the type of relationship they have with TW. These results can be used to tailor HIV prevention and care programs to best reach PTW, including those that focus on PrEP initiation and continuation.

## Methods

### Study design

We conducted a cross-sectional mixed methods study using an explanatory sequential design (QUAN→QUAL) to assess healthcare-seeking behavior and service preferences for HIV

testing and treatment among PTW and TW in Lima, Peru. This entailed conducting surveys using respondent driven sampling with PTW and TW, followed by interviews with a subset of survey participants. We analyzed surveys and interviews separately, then merged the findings, using the qualitative data to explain the quantitative survey results.

### Setting

The study was conducted in Lima, Peru, a city of 11 million people, including an estimated 22,500 TW [4, 6]. Recruitment of TW occurred at *casas trans*, informal living spaces for TW in Lima, primarily located in the central Cercado de Lima district. All study procedures were carried out by Epicentro, a community-based organization focused on LGBTQ+ health promotion, with over a decade of experience engaging with TW in Lima for research and HIV service delivery.

### Study population

Eligible individuals self-identified as a TW or reported to be a PTW, were at least 18 years old, and were literate in Spanish. TW were defined as individuals who reported being assigned male sex at birth but self-identified as female [20]. PTW were defined as individuals who reported having a sexual relationship with a TW in the last 6 months. While prior studies indicate that most PTW are cisgender men [3, 5, 11, 21], we did not exclude cisgender women or transgender men from the study.

### Sampling

We used a single round of respondent-driven sampling (RDS). We first recruited TW "seeds" to participate in a survey. After completing the survey, TW were invited to refer up to five of their recent sexual partners (PTW) to participate in the study by sending them a unique URL link to the same survey. There was no onward distribution of surveys from PTWs, yet we considered this a modified RDS approach as we retained the dual incentive structure typical of RDS by providing incentives for survey completion and an additional incentive to TW for successfully recruiting their partners [22]. All participants who completed the survey were given the option to leave their contact information if they were interested in participating in a follow-up interview. The study team attempted to contact all such participants, and in-depth interviews were scheduled with all who could be reached and remained interested.

### Data collection

We collected quantitative data from July 4–October 25, 2022 using a self-administered, web-based survey in REDCap, hosted at the Institute of Translational Health Sciences. TW participants could choose to complete the survey either on their personal cellphone or using a study tablet. Study staff assisted TW in loading the survey and were available for in-person technical support, as needed. PTW accessed the survey via a unique link sent by the referring TW "seed" and could complete the survey at their own convenience from any internet-capable personal device, without supervision from study staff. The survey links for PTW were programmed to expire after a single use, but there was no limit on the amount of time participants had to open and complete the survey. Participants provided their DNI number (8-digit unique Peruvian national identification) to prevent duplicate enrollment in the survey. A trained qualitative researcher (HS) conducted semi-structured interviews via phone from August–November 2022.

Both the survey and interview guide were developed in English and translated to Spanish by the study team (AL, CK) then edited for accuracy and local appropriateness by a native Peruvian Spanish-speaker on the team (HS). All interviews were audio recorded, transcribed in Spanish, then translated to English using professional translation services in Lima. Members of the study team (CK, HS) further reviewed translations to ensure accuracy prior to analyzing the data.

## Variables

The survey collected information on participant demographics, sexual behavior, HIV testing history, self-perceived HIV risk, experience with the healthcare system, and preferences related to either HIV testing or HIV treatment services, depending on the participant's self-reported HIV status (See S1 Annex). Questions on preferences for HIV testing and treatment covered three categories: location, convenience, and confidentiality/privacy. Within each category, questions assessed different factors that respondents rated on a Likert Scale, with 1 corresponding to "extremely important", 2 to "somewhat important," 3 to "neither important nor unimportant," 4 to "somewhat unimportant," and 5 to "extremely unimportant".

Interviews collected information on perceptions of individual HIV risk and interactions with the healthcare system, specifically in relation to HIV testing and linkage to HIV care (among those who identified as being HIV-positive). Interviews were designed to provide opportunities to explore topics covered in the survey in more depth.

Surveys and interviews also denoted the type of relationship PTWs had with the TW who referred them to the study (henceforth referred to as "partner type"): stable, casual, or transactional. PTW were classified as stable partners if they self-reported the referring TW to be their "stable partner/spouse". Casual partners were those who described their relationship with the referring TW as any of the following: "friends-with-benefits"; *punto* (colloquial term for a one-time partner); or someone with whom they occasionally had sex but not in exchange for money, goods or services. PTW were classified as transactional partners if they either sold sex to, or purchased sex from, the referring TW partner.

## Analysis

We conducted a primarily descriptive analysis of survey data. For questions using a Likert scale, we calculated means and frequencies; while presentation of frequencies is often best practice for Likert scales, calculation of means is also viewed as an acceptable approach to increase interpretability of results [23]. We assessed differences between TW and PTW responses to each question using two-sample t-tests (means) and chi-square (frequencies). To compare frequencies, we collapsed responses to the 5-point Likert scale questions and expressed these as dichotomous variables ("extremely important" vs. all other responses), given the unexpectedly high frequency of participants responding "extremely important" across all questions. Finally, we used ANOVA to assess for differences comparing the three different PTW partner types (stable, casual, transactional). We conducted all analyses in Stata 17.0 (College Station, TX).

Respondent driven sampling lacks a gold standard method to account for clustering in the analysis [24], but as our study used only one round of invitations (i.e., TW reaching out to their partners), we conducted a sensitivity analysis in R 4.1 (R Core Team, Vienna, Austria) using routine methods to adjust for clustering (packages "lme4", "lmerTest", "CR2", and "performance"). We did this for two sets of outcomes: difference in mean responses for TW and PTW preferences for (1) HIV testing and (2) HIV treatment. This analysis for HIV testing outcomes revealed very low ICC values, and there was no difference in significance for either HIV

testing or treatment outcomes. As sensitivity analyses did not alter interpretation of results, we did not conduct adjusted analyses for other outcomes.

After analyzing survey data, we conducted a rapid thematic analysis of the interview data in English using the framework method [25]. Two analysts (CK, DT) independently reviewed all transcripts, then met to discuss any differences in their reviews, come to consensus, and identify key themes that emerged from the data; Spanish-language transcripts were consulted if questions arose over meaning of the text. Key themes were reviewed and agreed upon by the researcher who conducted the interviews (HS). Data integration occurred through merging qualitative and quantitative results, with the qualitative findings used to explain and expound on key findings from the quantitative survey.

### Ethics, consent, and permissions

The study was approved by ethics boards at Via Libre (Lima, Peru) and University of Washington (Seattle, USA). Via Libre is a non-governmental organization dedicated to conducting research and providing services for people affected by HIV in Peru; its IRB is frequently used for review of HIV studies amongst sexual minorities occurring in Lima [26]. Participants provided written consent before engaging in surveys and verbal consent prior to engaging in interviews. All consent was provided in Spanish. Participants were compensated 25 Soles ($7 USD) for completion of surveys and 50 Soles ($13 USD) for interviews. TW seed participants received additional compensation of 20 Soles ($5 USD) for each referred PTW participant who completed the survey, up to a maximum of three referrals.

## Results

### Participants

A total of 234 surveys were completed and eligible for analysis: 69 from TW and 165 from PTW (Table 1). The majority of TW (n = 25) recruited 2 or 3 partners; 15 TW recruited no partners, while 12 TW recruited 5 partners. Amongst PTW, 13 (7.9%) identified as stable partners, 84 (50.9%) identified as casual partners, and 68 (41.2%) identified as transactional partners; 158 (95.8%) of PTW identified as cisgender men, 1 (0.6%) identified as a cisgender woman, 3 (1.8%) identified as non-binary or "other", and 3 (1.8%) identified as transgender men. TW and PTW respondents were similarly aged, but PTW reported higher university attendance, employment, and monthly incomes than TW. Twenty-four participants reported an existing HIV diagnosis (18 TW and 6 PTW).

We conducted follow-up interviews with 20 survey participants: four with TW and 16 with PTW. Of PTW interviewees, 5 were stable partners, 6 were casual partners, and 5 were transactional partners of the referring TW. All TW interviewees identified as sex workers, compared with one-quarter (n = 4) of PTW. Three-quarters of TW interviewees were living with HIV, compared to one-quarter of PTW interviewees.

### Quantitative findings

**Knowledge and experience with HIV testing/prevention.** Within the past year, nearly all TW had taken an HIV test, compared to three-quarters of PTW (Table 2). Both TW and PTW reported accessing HIV testing predominantly at Ministry of Health (MINSA) facilities, as well as state-sponsored sexually transmitted infection clinics (Centros de Referencia de Infecciones de Transmisión Sexual, or "CERITS"). Among PTW, the proportion who reported having ever previously taken an HIV test was similar across all partner types; however, stable PTW partners expressed higher interest in HIV testing compared to casual or transactional

**Table 1. Characteristics of study participants.**

| | All (n = 234) | TW[a] (n = 69) | PTW[b] (n = 165) |
|---|---|---|---|
| | n (%) | n (%) | n (%) |
| **Age, years** | | | |
| Median (IQR[c]) | 31 (26–38) | 28 (24–35) | 32 (28–39) |
| **Level of education** | | | |
| At least some primary | 25 (10.7) | 14 (20.3) | 11 (6.7) |
| Secondary | 160 (68.4) | 50 (72.5) | 110 (66.7) |
| Post-secondary | 49 (20.9) | 5 (7.2) | 44 (26.7) |
| **Nationality** | | | |
| Peruvian | 211 (90.2) | 64 (92.8) | 147 (89.1) |
| Venezuelan | 22 (9.4) | 4 (5.8) | 18 (10.9) |
| Other | 1 (0.4) | 1 (1.4) | 0 (0) |
| **Employment Status** | | | |
| Full time | 80 (34.2) | 4 (5.8) | 76 (46.1) |
| Part time | 85 (36.3) | 11 (15.9) | 74 (44.8) |
| Unemployed | 69 (29.5) | 54 (78.3) | 15 (9.1) |
| **Last Month's Income [PEN][d]** | | | |
| No income | 14 (6.0) | 5 (7.3) | 9 (5.5) |
| 1–749 | 36 (15.4) | 20 (29.0) | 16 (9.7) |
| 750–1000 | 115 (49.1) | 38 (55.1) | 77 (46.7) |
| 1001–1500 | 47 (20.1) | 3 (4.3) | 44 (26.7) |
| More than 1500 | 11 (4.7) | 0 (0.0) | 11 (6.7) |
| Prefer not to respond | 11 (4.7) | 3 (4.3) | 8 (4.8) |
| **Sexual Attraction** | | | |
| Cisgender men | 78 (33.3) | 63 (91.3) | 15 (9.1) |
| Cisgender women | 65 (27.8) | 5 (7.2) | 60 (36.4) |
| Transgender men | 7 (3.0) | 3 (4.3) | 4 (2.4) |
| Transgender women | 157 (67.1) | 7 (10.1) | 150 (90.9) |
| **Sex partners** | | | |
| Cisgender men | 80 (34.5) | 63 (91.3) | 17 (10.3) |
| Cisgender women | 83 (35.8) | 6 (8.7) | 77 (46.7) |
| Transgender men | 19 (8.1) | 7 (10.1) | 12 (7.3) |
| Transgender women | 164 (70.1) | 10 (14.5) | 154 (93.3) |
| **Self-reported HIV Status** | | | |
| Negative | 182 (77.8) | 43 (62.3) | 139 (84.2) |
| Unknown | 28 (12.0) | 8 (11.6) | 20 (12.1) |
| Positive | 24 (10.3) | 18 (26.1) | 6 (3.6) |

[a]Transgender women

[b]Partners of transgender women

[c]Interquartile Range; [d]Peruvian Nuevos Soles, the national currency of Peru. Exchange rate at the start of data collection (July 2022) was 1 PEN = $0.26 USD. The minimum monthly wage in Peru was 1025 PEN during the entire study period.

partners. Overall, satisfaction with HIV testing services was high among both TW and PTW. TW reported higher satisfaction than PTW in general for all clinic types, but more TW reported discrimination at health facilities (S1 Table). Notably, few TW, and even fewer PTW, had heard of pre-exposure prophylaxis (PrEP).

**Table 2. Knowledge of and experience with HIV testing and prevention amongst transgender women (TW) and partners of transgender women (PTW).**

| | TW (n = 63) | PTW (n = 164) | PTW Type | | |
| --- | --- | --- | --- | --- | --- |
| | | | Stable (n = 13) | Casual (n = 83) | Transactional (n = 68) |
| | n (%) | n (%) | n (%) | n (%) | n (%) |
| **KNOWLEDGE** | | | | | |
| **Ever taken HIV test[a]** | 61 (96.8) | 145 (88.4) | 11 (84.6) | 70 (84.3) | 64 (94.1) |
| **Interest in taking an HIV Test** | | | | | |
| Extremely interested | 30 (47.6) | 30 (18.3) | 8 (61.5) | 14 (16.9) | 8 (11.8) |
| Somewhat interested | 12 (19.0) | 115 (70.1) | 3 (23.1) | 56 (67.5) | 56 (82.4) |
| Neither interested nor uninterested | 1 (1.6) | 5 (3.0) | 0 | 3 (3.6) | 2 (2.9) |
| Somewhat uninterested | 0 | 7 (4.3) | 1 (7.7) | 5 (6.0) | 1 (1.5) |
| Extremely uninterested | 0 | 0 | 0 | 0 | 0 |
| *Not applicable (HIV+)[a]* | 18 (28.6) | 6 (3.7) | 0 | 5 (6.0) | 1 (1.5) |
| *Missing response* | 2 (3.2) | 1 (0.6) | 1 (7.7) | 0 | 0 |
| **Knowledge of PrEP** | | | | | |
| Never heard of PrEP | 40 (63.5) | 124 (75.6) | 5 (38.5) | 71 (85.5) | 48 (70.6) |
| **EXPERIENCE** | | | | | |
| **Facilities where they reported having ever HIV testing in the past[b]** | | | | | |
| Government hospital/clinic (EsSalud) | 7 (11.1) | 23 (14.0) | 4 (30.8) | 7 (8.4) | 12 (17.7) |
| Government hospital/clinic (MINSA) | 47 (74.6) | 122 (74.4) | 4 (30.8) | 60 (72.2) | 58 (85.3) |
| CERITS clinic | 48 (76.2) | 99 (60.4) | 4 (30.8) | 51 (61.4) | 44 (64.7) |
| Private hospital/clinic | 2 (3.2) | 21 (12.8) | 3 (23.1) | 12 (14.5) | 6 (8.8) |
| NGO (Epicentro, Via Libre) | 16 (25.4) | 20 (12.2) | 2 (15.4) | 14 (16.9) | 4 (5.9) |
| Community-based health campaign | 17 (27.0) | 8 (4.9) | 2 (15.4) | 6 (7.2) | 0 |
| Never tested | 0 | 2 (1.2) | 0 | 2 (2.4) | 0 |
| **At least somewhat satisfied with service received at HIV testing site[c]** | | | | | |
| Government hospital/clinic (EsSalud) | 6 (85.7) | 22 (95.7) | 4 (100) | 7 (100) | 11 (91.7) |
| Government hospital/clinic (MINSA) | 41 (87.2) | 115 (94.3) | 4 (100) | 55 (91.7) | 56 (96.6) |
| CERITSS clinic | 41 (85.4) | 96 (97.0) | 4 (100) | 49 (96.1) | 43 (97.7) |
| Private hospital/clinic | 2 (100) | 20 (95.2) | 3 (100) | 11 (91.7) | 6 (100) |
| NGO (Epicentro, Via Libre) | 15 (93.8) | 19 (95.0) | 2 (100) | 13 (92.9) | 4 (100) |
| Community-based health campaign | 16 (94.1) | 6 (75.0) | 2 (100) | 4 (66.7) | 0 |
| **Experienced discrimination or felt they were unfairly treated at HIV testing site^^** | | | | | |
| Government hospital/clinic (EsSalud) | 1 (14.3) | 4 (17.4) | 3 (75.0) | 0 | 1 (8.3) |
| Government hospital/clinic (MINSA) | 7 (14.9) | 0 (0) | 0 | 0 | 0 |
| CERITSS clinic | 6 (12.5) | 1 (1.0) | 1 (25.0) | 0 | 0 |
| Private hospital/clinic | 0 | 0 | 0 | 0 | 0 |
| NGO (Epicentro, Via Libre) | 0 | 0 | 0 | 0 | 0 |
| Community-based health campaign | 2 (11.8) | 0 | 0 | 0 | 0 |

Footnotes: [a]Missing responses were excluded from denominator; [b]Participants could have been tested at multiple sites, so these numbers will not sum to the total number of participants; [c]Among those who reported having ever received HIV testing at given site.

**Preferences for HIV testing services.** When asked to rate the importance of different options for how HIV testing services could be delivered (i.e., related to location, convenience, and confidentiality/privacy), the majority of TW and PTW rated most of these as "extremely" or "somewhat" important; no specific option was preeminent as the most (or least) important for either TW or PTW (Fig 1). Compared to PTW, TW rated more options as "extremely important", a difference which was statistically significant for most responses, whether

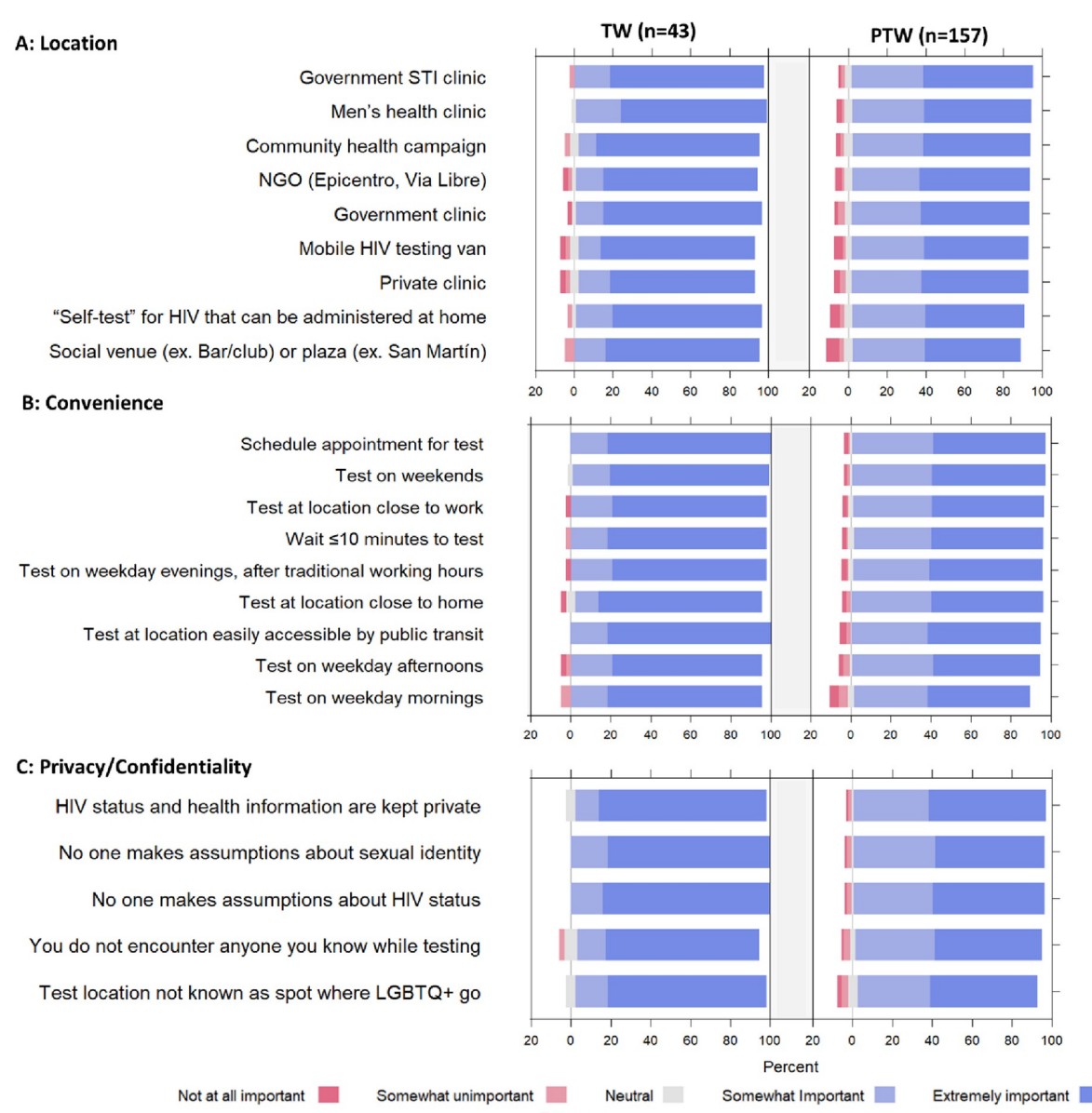

**Footnotes:**

[a]Ten eligible participants did not answer these questions (missing responses from 8 TW and 2 PTW), so n=200.

1A: Question stem read, "*If you were going to get an HIV test, how important would it be to test at ___*";
  Government STI clinic=CERITS; Government Clinic=EsSalud/MINSA, state-sponsored healthcare networks for workers and their families (EsSalud) or for general population as a safety net administered by Ministry of Health (MINSA)

1B: Question stem read, "*If you were going to get an HIV test, how important would it be to __*";

1C: Question stem read, "*If you were going to get an HIV test, how important would it be that ___*";
  When testing for HIV.

NB: Figure 1 was created using the "HH" package in R 4.1.0. Package citation: Heiberger RM (2022). *HH: Statistical Analysis and Data Display: Heiberger and Holland.*. R package version 3.1-49, https://CRAN.R-project.org/package=HH.

**Fig 1. Preferences for HIV testing among transgender women (TW) and partners of transgender women (PTW) who reported negative or unknown HIV status (n = 200)[a].** Bars to the left of '0' (color: red) indicate negative preferences (i.e., option is unimportant to respondent); bars to the right of '0' (color: blue) indicate positive responses (i.e., option is important to respondent); bars centered at '0' (color: grey) indicate neutral preferences. Together, all responses sum to 100%.

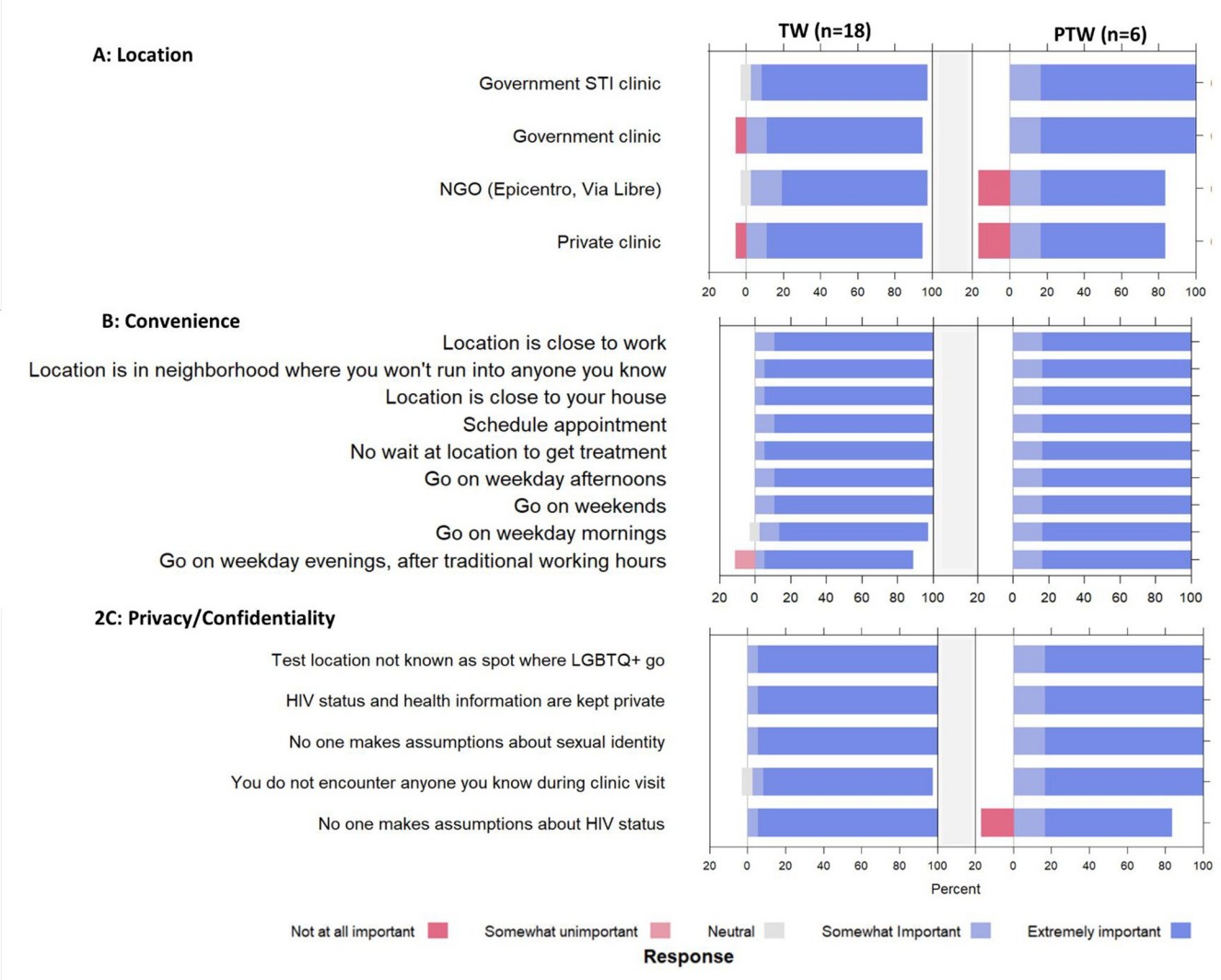

NB: Figure 2 was created using the "HH" package in R 4.1.0. Package citation: Heiberger RM (2022). *HH: Statistical Analysis and Data Display: Heiberger and Holland.*. R package version 3.1-49, https://CRAN.R-project.org/package=HH.

**Fig 2. Preferences for HIV treatment among transgender women (TW) and partners of transgender women (PTW) who reported being HIV-positive (n = 24).** Bars to the left of '0' (color: red) indicate negative preferences (i.e., option is unimportant to respondent); bars to the right of '0' (color: blue) indicate positive responses (i.e., option is important to respondent); bars centered at '0' (color: grey) indicate neutral preferences. Together, all responses sum to 100%.

comparing means (S2 Table) or frequencies (S3 Table). Among PTW subtypes, transactional partners were significantly most likely to rate any given option as "extremely important", while stable partners were the least likely to do so (S4 Table).

**Preferences for HIV treatment services.** Among participants who reported they were living with HIV, preferences for HIV treatment services demonstrated a slightly different pattern than what was observed for HIV testing. Almost all TW and PTW responded "extremely important" to all options presented in the survey, with no one option being clearly distinguishable as most or least important for either group (Fig 2). We observed no statistically significant

differences between TW and PTW regardless of whether the Likert scale means (S2 Table) or frequencies (S3 Table) were used for these comparisons.

## Qualitative findings

Qualitative interviews revealed four key themes that PTW and TW highlighted when discussing their experiences and preferences for HIV testing and treatment, which are described below. Additional details and supporting quotations are provided in S5 Table.

**Theme 1: Increase efficiency of HIV testing and treatment services.** The most common issue both TW and PTW interviewees mentioned when discussing HIV testing and treatment was long wait times, specifically excessive time required to secure appointments, long wait times during visits (whether scheduled or walk-in), and long wait times to receive test results. Several participants remarked that from start to finish, appointments for HIV testing could take 2 to 4 hours, which resulted in them missing work or other obligations. When asked about improvements they would recommend for HIV testing and treatment services, participants overwhelmingly reported wanting faster service.

> "*Generally, those are fast tests, they say: '40 minutes and the results are out.' But sometimes I think they exceed the capacity and there's too many people so you get to wait two-and-a-half-hours for the results and that's not the time they said.*" *(PTW-5, Transactional partner)*

**Theme 2: Focus on confidentiality, privacy, and discretion in HIV testing services.** Both TW and PTW highlighted the importance of HIV testing services being delivered in a discreet fashion. For example, interviewees often described mobile testing vans as a convenient and preferable option for HIV testing, but they remarked on the lack of privacy these offered. Notably, the desire for privacy specifically while standing in line outside of the mobile testing vans was mentioned more among the casual and transactional PTW, while stable partners were less likely to discuss this. While mobile HIV testing vans are unmarked, respondents explained that many people know their purpose. They often have long lines and wait times, and participants described how standing in line outside of such vans was frustrating, as it offered no discretion or privacy.

> "*There are things we don't like people to find out. As there in that square there are many comments from everyone. That's why I went to Patrucco [a CERITS clinic in central Lima]. . . Because one enters that mobile [testing van]. You know that this motive is for those results.*" *(PTW-11, Transactional partner)*

Other interviewees discussed being frustrated with situations they experienced in which providers demonstrated little respect for confidentiality, such as a patient's HIV status being discussed with others and sensitive questions being asked to patients while in a waiting room. Interviewees, specifically PTW, tried to report such instances to a manager or address the perpetrator directly, but there was no evidence of change.

> "*There was a guy leaving and it was my turn after him, so I got in there and there was the nurse with the doctor and they didn't even care that I was there, they started saying "Oh, the boy that left, he was positive" and stuff like that. I called them out saying that it wasn't proper for me that they were making those comments and it was worse that someone that had nothing to do with it was there, right? Because I didn't have the right to know if the guy was positive or not.*" *(PTW-5, Transactional partner)*

**Theme 3: Discrimination experienced in healthcare settings.** While less discussed than Theme 2, several interviewees described witnessing or experiencing blatant discrimination in healthcare settings. Discrimination was most noted against TW, but one PTW interviewee noted he experienced discrimination stemming from his relationships with TW. PTW who witnessed discrimination often reported addressing the perpetrators or talking to management, even if they did not know the TW who was discriminated against. Stable partners were particularly attuned to discrimination faced by TW.

"I've *seen that they keep them [TW] waiting, don't attend to them fast enough, and don't make things easier for them. . .. I mean, like they exclude them. They minimize them." (PTW-3, Stable partner)*

". . .*Sometimes they [healthcare workers] did it to laugh at you because sometimes they saw that you were a trans girl, they called you by the name that is on your ID. And you know there are people in the hospital; there are boys and ladies; everyone turns to look at you with your name; it's a shame for us." (TW-2)*

**Theme 4: Limited knowledge about PrEP.** The final theme that emerged was that there was limited knowledge about PrEP among interviewees, particularly amongst PTW. Most interviewees had never heard of, or knew very little about, PrEP. Several PTW expressed disbelief that such a medication existed. After being presented with a brief description of PrEP, most interviewees were very enthusiastic about taking PrEP and recommending it to friends, if it were to be available.

"*If I had known that five years ago, I would have been all in. If possible, the rest of my life would have continued with that. But, if nothing else, I believe it is a good thing that these pills are now available. You know what? I'm going to recommend that to my friends because a lot of my friends, when they lose their minds, [they] just have bareback sex and that's it" (PTW-1, Casual partner)*

This enthusiasm for PrEP was similar across all partner types, but reasoning for it differed. Stable partners related their enthusiasm for PrEP to the fact that either they or their TW partners were engaged in sex work, and that PrEP would offer protection in instances of failed barrier protection during work. On the other hand, casual and transactional partners framed their enthusiasm for PrEP in terms of personal safety when engaging in sexual intercourse with different partners.

## Integrated findings

We used qualitative findings to explain survey results (Table 3). While surveys did not distinguish any key preferences for HIV service delivery amongst TW or PTW, follow-up interviews revealed that convenience (specifically length of time it took to access testing or treatment services) and privacy (specifically confidentiality) were paramount for both TW and PTW when accessing HIV testing or treatment. Interviews also highlighted the challenge of discrimination against TW by healthcare providers in certain settings; while only a minority of survey participants reported experiencing discrimination in healthcare settings, those who did experience or witness discrimination had profoundly negative experiences. Furthermore, both surveys and interviews highlighted the limited knowledge TW and PTW had about PrEP, while interviews demonstrated the extreme interest these populations, particularly PTW, had in using PrEP, should it become available.

**Table 3. Joint display of quantitative and qualitative findings of transgender women (TW) and partners of transgender women (PTW) experiences and preferences for HIV services.**

| Category | Key Takeaways | Quantitative Findings | Qualitative Findings |
|---|---|---|---|
| Location | PTW are not currently accessing HIV testing through non-government sites frequented by TW (e.g., NGOs or community-based testing events that tend to target sexual and gender minorities). This could be an area to expand reach to PTW.<br><br>Additional sensitization at government-run sites is needed for healthcare providers to eliminate discrimination against TW and PTW. | The majority of HIV testing is accessed at government-run sites. More PTW accessed testing at private clinics, while more TW accessed testing through community-based campaigns at NGOs.<br><br>TW are more likely than PTW to experience discrimination when accessing HIV testing, with most discrimination reported occurring at government-run sites. | Interviewees did not focus on location preferences for testing/treatment. However, they did emphasis that gender-based discrimination in healthcare settings remains an issue. In general, TW were more likely to experience discrimination when accessing HIV testing, but PTW reported noticing when discrimination occurs towards TW in healthcare settings and were more likely than TW to confront healthcare workers when they witnessed or experienced discrimination. |
| Convenience | Improving efficiency of HIV testing and treatment may increase satisfaction and willingness to test for HIV among TW and PTW. | While efficiency is important to both groups, TW have stronger preferences than PTW for shorter wait times in clinic (not needing to wait more than 10 minutes) and being able to schedule an appointment for HIV testing. | Efficiency is an important aspect of HIV testing for transgender women and their partners. Specifically, long wait times during HIV testing visits were the most common complaint noted by interviewees. |
| Privacy/ Confidentiality | With regards to HIV testing and treatment, confidentiality is important to both TW and PTW and needs to be further emphasized by healthcare workers. | This issue rose to the top for majority of people. Over three-quarters of TW and over half of PTW thought this was extremely important. | Confidentiality, and specifically privacy, were mentioned as important factors by both TW and PTW. Several participants discussed witnessing breeches of confidentiality in healthcare settings, which contributes to their hesitancy in accessing HIV services. |
| Other (Education) | There is a need to provide increased education about PrEP to TW and their partners. | Most respondents (72.2% [n = 164/227]) had never heard of PrEP, with more PTW (75.6%) than TW (63.5%) being unaware of PrEP. Of note, stable PTW were much more likely to have heard of PrEP than casual or transactional partners. | Respondents, especially PTW, generally knew very little about PrEP, but were enthusiastic about learning more about the medication and recommending it to friends or taking it themselves once it becomes available. |

## Discussion

To the best of our knowledge, this is the first study to document the experiences and preferences that PTW have accessing HIV testing and treatment and how these compare to preferences of transgender women. While PTW and TW reported similar experiences and preferences for HIV services across the spectrum of service delivery assessed in our survey, there were notable differences between these two groups, namely PTW's lower overall interest in taking an HIV test, especially in non-traditional settings, and TW's experience of discrimination in healthcare settings. Moreover, our study expanded upon past studies that revealed PTW are not a monolithic group [7, 8], as PTW had different practices/preferences for HIV services varying by the type of relationship they had with transgender women. Interviews enabled us to better discern the relative importance of PTW preferences for HIV services and that they often aligned with needs expressed by TW—namely, that services should be offered more efficiently, privately, and free from discrimination. An unexpected finding of this study was the low level of PrEP knowledge, particularly amongst PTW.

We found it surprising that TW and PTW rated almost all potential options for HIV service delivery presented to them in the preference survey as important. This may be an artefact of how the participants were inclined to answer Likert scales (i.e., past research suggests there is a strong tendency for acquiescence when using Likert scales in surveys conducted in Latin America) [27]. While this limits the utility of the survey to guide the design of HIV service delivery to meet the needs of PTWs, the survey did offer valuable insight into how PTW experiences and preferences compare to those of TW. Most notably, similarities in responses between PTW and TW suggest that HIV services targeting PTW may be appropriate to offer

alongside those targeting TW. Differences in experiences highlighted by the survey, such as PTW's lesser interest in taking an HIV test and more limited knowledge of PrEP, may highlight how PTW have likely been missed by current HIV prevention outreach efforts. Other slight differences noted between TW and PTW may be due to the power differential between these populations. Power differences between TW and their partners have been well-noted in the literature, largely due to cis males being in the greatest position of power in many societies [1, 3, 8, 11]. This power differential between PTW and TW was observed in our survey findings, as PTW appeared to be of higher socioeconomic status than TW, evidenced by PTW generally having higher levels of education, employment, and income than their TW partners. Because PTWs have more power in society than TWs, we hypothesize that PTWs may be able to be more flexible in accessing HIV services and thus feel less strongly than TW about how, when, or where services are provided.

Another noteworthy finding from our study is that PTW are not a monolithic group and, as such, HIV prevention outreach and services may need to be tailored to meet each subgroup's specific needs. We hypothesize that differences observed in responses by PTW partner type could be explained by differences in their closeness to TW. For example, transactional sexual partners may have significantly stronger preferences than other PTW partner types on how HIV testing occurs because their relationships with TW are most likely to be clandestine, and thus they feel a greater need for confidentiality and convenience [8]. In contrast, the relational bonds between TW and their committed partners may provide a sense of security and community that leads them prioritize HIV testing/treatment, regardless of convenience and confidentiality.

Ultimately, the qualitative interviews in this study were most useful in highlighting how HIV services could be tailored to better reach PTW in Lima. The overarching principles important to PTW also rang true for TW, suggesting a focus on these core elements could improve reach to both PTW and TW: implementing procedures to reduce wait times for testing and results, and strengthening sensitization for healthcare workers on patient privacy and transgender care.

These suggestions speak to the specific challenges of accessing and interacting with the local health system in Lima and align with key challenges noted in the literature, namely an under resourced primary health care system and the need to address stigma surrounding HIV in the healthcare setting [28–31]. Interestingly, interviews also highlighted a core difference in how PTW may wish to interact with HIV services, in comparison to TW. PTW, specifically those not in stable relationships with TW, voiced concern over the lack of privacy offered by some non-traditional testing events, namely on-site, mobile HIV testing units; such concerns were not voiced by TW in our study. While this finding is based on a small number of interviews, it echoes findings in past qualitative research conducted in Peru [7], suggesting that while PTW and TW have similar preferences for many aspects of HIV testing, PTW, specifically those not in stable relationships with TW, may be better reached through traditional testing venues.

Finally, we were surprised to discover the limited knowledge that both TW and PTW had about PrEP. While limited PrEP availability and low rates of persistent PrEP use among key populations in Peru have been well-documented [32, 33], the low level of PrEP awareness we found among TW and PTW in this study is alarming. Although the World Health Organization has recommended PrEP for at-risk populations since 2015 [34], PrEP was only adopted as part of Peru's national HIV prevention guidelines in June 2023 [19]. As such, there is now an urgent need to tailor PrEP education and demand creation to at-risk populations in Peru, including TW and PTW. While doing so, our research, building upon past research, suggests we must recognize that PTW may be quite heterogenous in their needs, and their needs likely differ from those of gay-identifying MSM and, to a lesser extent, from TW [7–10, 14]. The

overwhelming enthusiasm that participants expressed towards PrEP offers encouragement that PrEP uptake could be widespread once it is accessible in Peru, but concerted efforts are likely needed to ensure adherence [32, 33].

## Limitations

This study was subject to several limitations. Firstly, we relied on convenience sampling to recruit initial TW participants for the survey, as well as to select participants for qualitative interviews. TW were recruited only at *casas trans*, which may limit the generalizability of our findings, as these individuals may not be representative of the broader TW population in Lima. Secondly, we did not adjust for clustering for all comparisons between PTW and TW as our sensitivity analysis of mean preferences for HIV testing and treatment between PTW and TW suggests adjusted analyses would not change interpretation of the results. Furthermore for "testing" outcomes, ICC values were very small, suggesting limited impact of clustering; for "treatment" outcomes, the impact of clustering appeared to be greater, but there were too few observations for any results to be significant in the adjusted or unadjusted analysis. Thirdly, limitations of Likert scales are well-recognized; responses were skewed towards acquiescence ("extremely" or "somewhat" important). In the future, a ranked scale could be better to identify the importance of each factor to participants [27]. Our qualitative findings helped elucidate such distinctions in this study, but additional research—namely a discrete choice experiment—is needed to better ascertain the order and relative magnitudes of importance among these preferences. Finally, due to the few people living with HIV included in this study, our findings on preferences for HIV treatment services should be interpreted with caution.

## Conclusion

This mixed methods study found many aspects of HIV services were important to PTW and generally aligned with TW preferences, but that PTW are a heterogeneous group whose individual preferences may vary based on their relationship with TW. Improving efficiency (e.g., reducing waiting time), increasing privacy while accessing HIV services, and eliminating discrimination based on gender and sexual preferences in healthcare settings should be prioritized to better reach both PTW and TW. There is also a need to increase awareness about PrEP, particularly amongst PTW. These results should be considered when designing HIV testing and prevention programs for PTWs. Moving forward, a discrete choice experiment amongst PTW should be conducted to ascertain the relative importance of aspects considered in this study and to understand how PTW make decisions about accessing HIV services.

## Supporting information

**S1 Annex. Survey questions administered to TW and PTW.**
(DOCX)

**S1 Table. Satisfaction with HIV testing among TW and PTW by facility type.**
(DOCX)

**S2 Table. Service preference means for HIV testing and treatment services (TW vs PTW).**
(DOCX)

**S3 Table. Frequency of "extremely important" responses (TW vs PTW).**
(DOCX)

**S4 Table. Service preference means for HIV testing services by PTW type.**
(DOCX)

**S5 Table. Themes regarding service preference from qualitative interviews.**
(DOCX)

## Author Contributions

**Conceptualization:** Deanna Tollefson, Hugo Sanchez, Ann Duerr.

**Formal analysis:** Claudia Kazmirak, Deanna Tollefson.

**Funding acquisition:** Claudia Kazmirak, Alexander Lankowski, Ann Duerr.

**Investigation:** Claudia Kazmirak, Alexander Lankowski, Hugo Sanchez, Ivan Gonzales, Dianne Espinoza.

**Methodology:** Claudia Kazmirak, Deanna Tollefson.

**Project administration:** Claudia Kazmirak, Alexander Lankowski, Hugo Sanchez.

**Resources:** Hugo Sanchez.

**Supervision:** Alexander Lankowski, Ann Duerr.

**Visualization:** Deanna Tollefson.

**Writing – original draft:** Claudia Kazmirak, Deanna Tollefson.

**Writing – review & editing:** Deanna Tollefson, Alexander Lankowski, Hugo Sanchez, Ivan Gonzales, Dianne Espinoza, Ann Duerr.

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
