## [Decision Letter · Decision Letter 0]

18 Oct 2023

PONE-D-23-29942Practices and preferences for HIV testing and treatment services amongst partners of transgender women in Lima, Peru: an exploratory, mixed methods studyPLOS ONE

Dear Dr. Tollefson,  Upon reviewing the findings, it is evident that there is significance. In alignment with the reviewers' comments, I concur that major revisions are necessary.,

Thank you for submitting your manuscript to PLOS ONE. After careful consideration, we feel that it has merit but does not fully meet PLOS ONE’s publication criteria as it currently stands. Therefore, we invite you to submit a revised version of the manuscript that addresses the points raised during the review process.

We look forward to receiving your revised manuscript.

Kind regards,

Alex Siu Wing Chan

Academic Editor

PLOS ONE

Journal Requirements:

Research reported in this publication was supported by the Fogarty International Center of the National Institutes of Health under grants 5D43TW009345-10 and 2D43TW009345-11 awarded to the Northern Pacific Global Health Fellows Program. We acknowledge support related to the use of ITHS REDCap to carry out this study (UL1 TR002319, KL2 TR002317, and TL1 TR002318 from NCATS/NIH). The content is solely the responsibility of the authors and does not represent the views of the National Institutes of Health.

Additional Editor Comments:

Thank you for submitting this intriguing study to PLOS ONE. Upon reviewing the findings, it is evident that there is significance. In alignment with the reviewers' comments, I concur that major revisions are necessary.

Reviewers' comments:

Reviewer's Responses to Questions

**Comments to the Author**

1. Is the manuscript technically sound, and do the data support the conclusions?

Reviewer #1: Yes

Reviewer #2: Yes

Reviewer #3: Yes

2. Has the statistical analysis been performed appropriately and rigorously? 

Reviewer #1: Yes

Reviewer #2: Yes

Reviewer #3: Yes

3. Have the authors made all data underlying the findings in their manuscript fully available?

Reviewer #1: Yes

Reviewer #2: Yes

Reviewer #3: Yes

4. Is the manuscript presented in an intelligible fashion and written in standard English?

Reviewer #1: Yes

Reviewer #2: Yes

Reviewer #3: Yes

5. Review Comments to the Author

Reviewer #1: Thank you for the opportunity to review, “Practices and preferences for HIV testing and treatment services amongst partners of transgender women in Lima, Peru: an exploratory, mixed methods study.” The authors present results of a mixed-methods survey of transwomen (TW) and their sexual partners (PTW) in Lima, Peru regarding their preferences and practices for HIV testing and treatment. The results are interesting and important, presenting a relatively large sample of PTW (a population that has been understudied to date) addressing differences in attitudes towards and use of HIV prevention and testing services. The analysis is well done and the manuscript well written. There are a few points for clarification or elaboration prior to publication:

1) The authors describe their sampling method as Respondent Driven Sampling (RDS) but do not use any of the analytic methods inherent to RDS, and the single-wave of recruitment from seeds selected through convenience sampling is not consistent with RDS methods. It may be preferable to describe recruitment as peer recruitment or snowball sampling to avoid methodological disputes over terminology.

2) In the abstract, it would help to specify the numbers of TW and PTW recruited for the study.

3) The Background is brief and does not provide any context for understanding why the authors chose to address differences among PTW according to their partnership status or partner type with TW. Given that Jess Long’s papers, cited in the manuscript, provide a great amount of detail on this topic, it would help to have some discussion of this work in the Background to set the stage for the rest of the analysis.

4) In the Methods, participants are required to have been “literate in Spanish.” Does this comment mean that participants who were unable to read or write were excluded from participation?

5) What were the recruitment patterns for TW seeds and the PTWs (i.e., were a few seeds responsible for a great majority of the PTWs, what was the distribution of PTW characteristics in relation to TW seeds, etc.)?

6) What were the demographic and other characteristics of the subset of participants in the qualitative interviews?

7) Please specify in the Methods how “Stable partners” were defined in the survey (similar to the explanations provided for casual and transactional partners).

8) Was the qualitative data analyzed in the original Spanish or in English translation?

9) Why was Via Libre’s IRB used for this study (a brief explanation would be useful)?

10) The right side of Table 2 is cut off in the current formatting.

11) In the Discussion, the authors suggest that power differentials in the relationships between TW and PTW are partly due to the higher SES of PTW. However, in this sample, over 95% of all participants (and 94% of PTW) report earning less than $400 USD/month. It is more likely that other factors, like gendered power differentials between cis-men and transwomen, and the high frequency of sex workers in the TW sample, play a greater role in defining power differentials in these relationships.

12) The authors seem to gloss over a potentially fascinating observation in their qualitative data, where PTW complained about on-site HIV testing with vans or mobile units as these can increase visibility of individuals waiting outside the van for testing, and some stating that they prefer to attend traditional testing venues as they provide greater privacy and discretion, while the TW interviewed did not have similar complaints. This observation could have critical implications for how HIV testing and treatment is provided to these different groups, and merits greater discussion.

Reviewer #2: In article ‘Practices and preferences for HIV testing and treatment services amongst partners of transgender women in Lima, Peru: an exploratory, mixed methods study’ submitted for review at PLOS One, the authors conducted a cross-sectional mixed methods study among Partners of Transgender Women (PTW) and Transgender Women (TW) in Lima, Peru using an explanatory sequential design where online surveys were administered followed by interviews. Based on the findings, practices and perspectives on HIV testing and treatment were compared quantitatively and qualitatively between PTW and TW participants. Firstly, I want to congratulate the authors for targeting an extremely hard to reach population (PTW) for some very timely research, since the global drive to end the HIV epidemic would not be successful without HIV prevention and care access for all hidden, vulnerable, hard to reach (and understand) populations like the TW and PTW. The article is well written, easy to follow, and the conclusions seem generally plausible given the results of the statistical analyses. However, I do have a few comments:

Major Comments

1. The authors mentioned this as one of their limitations, but I wonder how much of the perceived differences in demographic characteristics between TW and PTW are driven by the difference in recruitment of the respective groups. TW were recruited from clinics and can potentially be more vulnerable than the average TW population in Lima, and PTW were recruited by the TW seeds, by virtue of the definition, were probably from a more general pool of PTWs in Lima.

Do the researchers have any data to support or refute that claim? For example, one metric might be HIV prevalence among recruited TW versus the general TW population, which seems to be similar in this case (30% in GP vs 26.1-37.7% in the study). Another important factor might be sex work status. Sex work is a known risk factor for HIV and other STIs as well and is known to be correlated with lower income and lower education. Can the researchers track the sex work status of all survey TW participants, and compare them with the TW population average in Lima? Any other metric to use perhaps?

2. Another limitation that the authors mention about the statistical analysis is the following:

‘Secondly, we did not adjust for clustering present from respondent driven sampling because this was an exploratory study and there is no gold standard for analysis for this sampling method.’

The authors are probably correct in pointing out that there is no gold standard method for analysis of RDS data in the literature as of yet, however, note that a typical RDS has multiple waves of recruitment with a few initial seeds, which makes statistical modeling quite challenging because of the hierarchy of the different waves. However, the study design pursued here only employed one wave of RDS, and thus the ensuing data can be conceived as clustered, where each cluster consists of a TW seed and the PTWs they recruited. The authors should at least look whether such clustering exists (with respect to outcomes in Tables S3 and S4), for example, one can estimate the intra-cluster correlation coefficient (ICC). If indeed the ICC is high, one can pursue adjusted t-tests/chi-square tests for clustered data or regression approaches like LME to see if the inference is any different from the unadjusted analyses.

Minor Comments

1. Page 11, line 180, Table 1 – Median and IQR are probably better metrics for ‘Age’ than mean and SD.

2. Page 13, line 197, Table 2 – Not fully visible for review. A third of the table seemed to be cut-off, probably that page needs to be in landscape format.

Reviewer #3: The authors evaluated practices and preferences for HIV services in partners of TW, a highly at-risk hidden population.

The paper is clearly written and only minor edits are needed.

For readers who are not familiar with the Peruvian health system and current standard of care, I consider the paper could benefit from the following changes/additions:

-The authors described in the discussion that PrEP was implemented in Peru in June 2023. Since this question arose at the time of the results, I suggest moving this explanation to the introduction.

-On page 3, line 17, the authors mentioned “non-traditional HIV testing options”. I suggest adding in the introduction a short explanation about the standard of care for HIV testing in Peru and clarifying what are the “non-traditional HIV testing options” available.

6. PLOS authors have the option to publish the peer review history of their article (what does this mean?). If published, this will include your full peer review and any attached files.

Reviewer #1: **Yes: **Jesse Clark

Reviewer #2: No

Reviewer #3: No

---

## [Author Response · Author response to Decision Letter 0]

21 Feb 2024

Response to reviewers is included in the attached MS Word document.

---

## [Decision Letter · Decision Letter 1]

1 Apr 2024

PONE-D-23-29942R1Practices and preferences for HIV testing and treatment services amongst partners of transgender women in Lima, Peru: an exploratory, mixed methods studyPLOS ONE

Dear Dr. Tollefson,

Thank you for submitting your manuscript to PLOS ONE. After careful consideration, we feel that it has merit but does not fully meet PLOS ONE’s publication criteria as it currently stands. Therefore, we invite you to submit a revised version of the manuscript that addresses the points raised during the review process.

**ACADEMIC EDITOR:**

While the reviewers have already provided their comments, here are my suggestions and concerns:

1. Choose either an English or Spanish abstract.

2. Even though you've discussed the transgender situation globally, please clarify the correlation between your topic and the research question in the introduction.

3. I suggest revising the aim of this study to articulate how this research will make a significant contribution.

4. It appears you are using a mixed-methods approach; please be more specific in the data analysis section.

5. The table 1 presents a detailed breakdown of the characteristics of study participants, including age, education level, nationality, employment status, income, sexual attraction, number of sex partners, and self-reported HIV status, differentiated between TW (transgender women, n=69) and PTW (presumably non-transgender participants, n=165). regarding Education and Employment: Could the authors delve into the factors contributing to the educational and employment disparities observed, particularly the barriers TW face in these areas?

6. How does the cultural and national context influence the study's findings, especially regarding social support systems and healthcare access?

7. The authors are invited to explore how the sexual attraction and partner preferences among TW and PTW inform our understanding of HIV risk and prevention needs within these communities.

8. Given the marked HIV status disparity, what specific, actionable recommendations do the authors propose to mitigate this risk among TW?

Please submit your revised manuscript by May 16 2024 11:59PM.   If you will need more time than this to complete your revisions, please reply to this message or contact the journal office at plosone@plos.org. Please include the following items when submitting your revised manuscript:A rebuttal letter that responds to each point raised by the academic editor and reviewer(s). You should upload this letter as a separate file labeled 'Response to Reviewers'.A marked-up copy of your manuscript that highlights changes made to the original version. You should upload this as a separate file labeled 'Revised Manuscript with Track Changes'.An unmarked version of your revised paper without tracked changes. You should upload this as a separate file labeled 'Manuscript'.If applicable, we recommend that you deposit your laboratory protocols in protocols.io to enhance the reproducibility of your results. Protocols.io assigns your protocol its own identifier (DOI) so that it can be cited independently in the future. For instructions see: https://journals.plos.org/plosone/s/submission-guidelines#loc-laboratory-protocols. Additionally, PLOS ONE offers an option for publishing peer-reviewed Lab Protocol articles, which describe protocols hosted on protocols.io. Read more information on sharing protocols at https://plos.org/protocols?utm_medium=editorial-email&utm_source=authorletters&utm_campaign=protocols.

We look forward to receiving your revised manuscript.

Kind regards,

Alex Siu Wing Chan

Academic Editor

PLOS ONE

Journal Requirements:

Reviewers' comments:

Reviewer's Responses to Questions

**Comments to the Author**

1. If the authors have adequately addressed your comments raised in a previous round of review and you feel that this manuscript is now acceptable for publication, you may indicate that here to bypass the “Comments to the Author” section, enter your conflict of interest statement in the “Confidential to Editor” section, and submit your "Accept" recommendation.

Reviewer #1: All comments have been addressed

Reviewer #2: All comments have been addressed

Reviewer #3: All comments have been addressed

2. Is the manuscript technically sound, and do the data support the conclusions?

Reviewer #1: Yes

Reviewer #2: Yes

Reviewer #3: Yes

3. Has the statistical analysis been performed appropriately and rigorously? 

Reviewer #1: Yes

Reviewer #2: Yes

Reviewer #3: Yes

4. Have the authors made all data underlying the findings in their manuscript fully available?

Reviewer #1: Yes

Reviewer #2: Yes

Reviewer #3: Yes

5. Is the manuscript presented in an intelligible fashion and written in standard English?

Reviewer #1: Yes

Reviewer #2: Yes

Reviewer #3: Yes

6. Review Comments to the Author

Reviewer #1: (No Response)

Reviewer #2: The authors have done an excellent job addressing all the comments left by the Reviewers. I just have one minor comment that is mainly about consistency in presentation of results.

1(i) In Supplementary Tables S3 and S5 (and possibly others), the authors have presented standard deviations (I presume for the data distribution), however, given they present results from hypothesis tests conducted on the means, the more relevant metric would be standard errors for these parameters. Can you present standard errors instead of SDs (or in addition to the SDs) in the Tables?

1(ii) Additionally, to be consistent, can you please add the cluster-adjusted standard errors in Supplementary Table S3-2?

Reviewer #3: the authors have addressed most reviewers comments, corrected tables format and the manuscript is now clear

7. PLOS authors have the option to publish the peer review history of their article (what does this mean?). If published, this will include your full peer review and any attached files.

Reviewer #1: No

Reviewer #2: No

Reviewer #3: No

---

## [Author Response · Author response to Decision Letter 1]

22 Apr 2024

We have provided the full response to the editor and reviewer #2 in an attached Microsoft Word document. In addition, we have pasted our response below.

Response To Academic Editor

1. Choose either an English or Spanish abstract.

Response: Thank you for your comment. We have removed the Spanish abstract from the submission. 

2. Even though you've discussed the transgender situation globally, please clarify the correlation between your topic and the research question in the introduction.

Response: Thank you for your observation and this suggestion. We have modified the wording in the introduction section to better highlight the connection between the topic (HIV testing and prevention services amongst PTW) and the research question at hand. Most prominently, we have adjusted the first sentence of the paper to include mention of the partners of transgender women to emphasize to the readers that partners of transgender women (PTW), not transgender women, are the focus of this paper. 

The research question for this paper is: What are the practices and preferences for HIV testing and treatment amongst the sexual partners of transgender women in Lima, Peru? This is included in statement form in the manuscript in lines 49-52, where we define the goals of this study: “We therefore conducted this study to describe the practices and preferences that PTW in Peru have for HIV testing and treatment services and to determine how such practices and preferences (1) compare to those of TW and (2) compare amongst PTW based on the type of relationship they have with TW.”

We have described the importance of understanding the preference of HIV testing amongst partners of transgender women in lines 25-48 in the Introduction section. In particular, we have described the need to examine this issue in Peru in lines 34-48. 

3. I suggest revising the aim of this study to articulate how this research will make a significant contribution.

Response: Thank you for this suggestion. We have modified our statement about how this research will make a significant contribution in lines 52-53. This now reads: “These results can be used to tailor HIV prevention and care programs to best reach PTW, including those that focus on PrEP initiation and continuation.”

In addition, we have added a sentence to the conclusion to highlight the importance of this work (see line 365): “These results should be considered when designing HIV testing and prevention programs for PTWs.”

4. It appears you are using a mixed-methods approach; please be more specific in the data analysis section.

Response. Thank you for this comment. 

▪ Per best practice, we have included our study design in the title of this study (“an exploratory, mixed methods study”). 

▪ We have followed best practices to describe the mixed methods design in this study in the first paragraph of the methods section (under study design, see lines 57-61). Here we state that we used an explanatory sequential design (QUAN�QUAL). Per best practices for reporting qualitative data, we have now also clarified that data integration happened through merging of the data: “We analyzed surveys and interviews separately, then merged the findings, using the qualitative data to explain the quantitative survey results.” (Lines 60-61).

▪ We have also clarified this in the ‘data analysis’ section of the results, specifically calling out “merging” as the form of data integration. This section in our manuscript now reads: “Data integration occurred through merging qualitative and quantitative results, with the qualitative findings used to explain and expound on key findings from the quantitative survey.” (Lines 144-145)

As reference, below is a document that describes some of the practices for mixed methods research: 

NIH Office of Behavioral and Social Sciences. (2018). Best practices for mixed methods research in the health sciences (2nd ed). Bethesda: National Institutes of Health. Accessed via https://implementationscience-gacd.org/wp-content/uploads/2020/11/Best-Practices-for-Mixed-Methods-Research-in-the-Health-Sciences-2018-01-25-1.pdf

5. The table 1 presents a detailed breakdown of the characteristics of study participants, including age, education level, nationality, employment status, income, sexual attraction, number of sex partners, and self-reported HIV status, differentiated between TW (transgender women, n=69) and PTW (presumably non-transgender participants, n=165). regarding Education and Employment: Could the authors delve into the factors contributing to the educational and employment disparities observed, particularly the barriers TW face in these areas?

Response: Thank you for this thoughtful comment. We agree that it is interesting to consider the differences between TW and PTW. Unfortunately, delving into the specifics of these differences goes beyond the scope of our paper and involves much speculation. While we do not think it is the place of this study to specifically explore this difference, we do provide some speculation as to why these differences exist: the power differential between cis male PTW and TW (see lines 300-307). We use our qualitative findings, and past studies, to support this hypothesis. 

We have rephrased this discussion in lines 301-305 to read as follows: “Power differences between TW and their partners have been well-noted in the literature, largely due to cis males being in the greatest position of power in many societies [1,3,7,10]. This power differential between largely cis-gender male PTW and TW was observed in our survey findings, as PTW appeared to be of higher socioeconomic status than TW, evidenced by PTW generally having higher levels of education, employment, and income than their TW partners.” 

6. How does the cultural and national context influence the study's findings, especially regarding social support systems and healthcare access?

Response: Thank you for this question. We have provided discussion on how the local cultural context influences findings in the discussion section. For example: 

▪ In the second paragraph of the discussion (Lines 292-307) ,we discuss (1) how local culture could have made the Likert scales used in the study ineffective, thus limiting what we could conclude from our survey, and (2) the power differential that likely exists between PTW and TW due to the local context, and how this could affect their access to health care. 

▪ In the fourth paragraph of the discussion, we discuss specific recommendations for improving access to HIV treatment services: decreasing wait time and strengthening sensitization for healthcare workers on patient privacy and transgender care. These are both directly related to the local context of how HIV testing services are delivered in Peru. We have added new additional language (and additional citations) into the text to make note of the influence local context has on this issue (Lines 322-324). 

“These suggestions speak to the specific challenges of accessing and interacting with the local health system in Lima and align with key challenges noted in the literature, namely an under resourced primary health care system and the need to address stigma surrounding HIV, including in the healthcare setting (28–31).”

7. The authors are invited to explore how the sexual attraction and partner preferences among TW and PTW inform our understanding of HIV risk and prevention needs within these communities.

Response: Thank you for this suggestion. We agree that this is an important topic of research. Unfortunately, the discussion of partner preference and risk is outside the scope of our current study, which seeks to describe the practices and preferences that PTW in Peru have for HIV testing and treatment services. 

However, we have addressed this in our past research, which we cite in this paper. For more information on this topic, please see these other papers published by our research group (which we have also referred to in this manuscript): 

Long JE, Montaño M, Sanchez H, Huerta L, Calderón Garcia D, Lama JR, et al. Self-Identity, Beliefs, and Behavior Among Men Who Have Sex with Transgender Women: Implications for HIV Research and Interventions. Arch Sex Behav. 2021; 50(7):3287–95. 

Long JE, Ulrich A, White E, Dasgupta S, Cabello R, Sanchez H, et al. Characterizing Men Who Have Sex with Transgender Women in Lima, Peru: Sexual Behavior and Partnership Profiles. AIDS Behav. 2020; 24(3):914–24. 

Long JE, Sanchez H, Dasgupta S, Huerta L, Calderón Garcia D, Lama JR, et al. Exploring HIV risk behavior and sexual/gender identities among transgender women and their sexual partners in Peru using respondent-driven sampling. AIDS Care. 2022;34(9):1187–95. 

Long JE, Sanchez H, Dasgupta S, Huerta L, Garcia DC, Lama JR, et al. Self-Reported Knowledge of HIV Status Among Cisgender Male Sex Partners of Transgender Women in Lima, Peru. J Acquir Immune Defic Syndr. 2022 May 1;90(1):1–5. 

Long JE, Tordoff DM, Reisner SL, Dasgupta S, Mayer KH, Mullins JI, et al. HIV transmission patterns among transgender women, their cisgender male partners, and cisgender MSM in Lima, Peru: A molecular epidemiologic and phylodynamic analysis. Lancet Reg Health Am. 2022; 6.

8. Given the marked HIV status disparity, what specific, actionable recommendations do the authors propose to mitigate this risk among TW?

Response: We appreciate this question. The goal of this paper was to describe the practices and preferences for HIV prevention amongst partners of transgender women. TW were not the focus of our paper. We did not conduct HIV testing in this study, or compare the prevalence of HIV amongst TW, PTW, or the broader population, so we believe proposing actionable research to mitigate the risk of HIV amongst TW is outside the scope of this paper. 

Response To Reviewer

Reviewer #2: The authors have done an excellent job addressing all the comments left by the Reviewers. I just have one minor comment that is mainly about consistency in presentation of results.

▪ In Supplementary Tables S3 and S5 (and possibly others), the authors have presented standard deviations (I presume for the data distribution), however, given they present results from hypothesis tests conducted on the means, the more relevant metric would be standard errors for these parameters. Can you present standard errors instead of SDs (or in addition to the SDs) in the Tables?

▪ Additionally, to be consistent, can you please add the cluster-adjusted standard errors in Supplementary Table S3-2?

Response: 

Thank you for this note. This was a descriptive study, so we are describing the sample population in all the data presented in the supplemental tables. We are not seeking to make inference on the overall population. 

As standard deviations are used to describe the sample, while standard errors are used when making inference on the broader population, we believe that inclusion of standard deviation, not standard errors, is the most appropriate choice for these tables. 

We now uniformly presented means and standard deviations in the relevant supplemental tables (S3 & S5). 

Relevant sources: 

Nagele P. Misuse of standard error of the mean (SEM) when reporting variability of a sample. A critical evaluation of four anaesthesia journals. Br J Anaesth. 2003. 90(4): 514-6. doi: 10.1093/bja/aeg087

Kim H-Y. Statistical notes for clinical researchers: Understanding standard deviations and standard errors. Restor Dent Endod. 2013. 38(4): 263-265. doi: 10.5395/rde.2013.38.4.263

---

## [Decision Letter · Decision Letter 2]

26 Jun 2024

Practices and preferences for HIV testing and treatment services amongst partners of transgender women in Lima, Peru: an exploratory, mixed methods study

PONE-D-23-29942R2

Dear Dr. Tollefson,

We’re pleased to inform you that your manuscript has been judged scientifically suitable for publication and will be formally accepted for publication once it meets all outstanding technical requirements.

Kind regards,

Miquel Vall-llosera Camps

Senior Staff Editor

PLOS ONE

Reviewers' comments:

Reviewer's Responses to Questions

**Comments to the Author**

1. If the authors have adequately addressed your comments raised in a previous round of review and you feel that this manuscript is now acceptable for publication, you may indicate that here to bypass the “Comments to the Author” section, enter your conflict of interest statement in the “Confidential to Editor” section, and submit your "Accept" recommendation.

Reviewer #1: All comments have been addressed

Reviewer #2: All comments have been addressed

Reviewer #3: All comments have been addressed

2. Is the manuscript technically sound, and do the data support the conclusions?

Reviewer #1: Yes

Reviewer #2: Yes

Reviewer #3: Yes

3. Has the statistical analysis been performed appropriately and rigorously? 

Reviewer #1: Yes

Reviewer #2: Yes

Reviewer #3: Yes

4. Have the authors made all data underlying the findings in their manuscript fully available?

Reviewer #1: Yes

Reviewer #2: Yes

Reviewer #3: Yes

5. Is the manuscript presented in an intelligible fashion and written in standard English?

Reviewer #1: Yes

Reviewer #2: Yes

Reviewer #3: Yes

6. Review Comments to the Author

Reviewer #1: (No Response)

Reviewer #2: (No Response)

Reviewer #3: Authors addressed comments for the first revision round. No additional comments for this second review

7. PLOS authors have the option to publish the peer review history of their article (what does this mean?). If published, this will include your full peer review and any attached files.

Reviewer #1: No

Reviewer #2: No

Reviewer #3: No

---

## [Editor Report · Acceptance letter]

28 Jun 2024

PONE-D-23-29942R2 

PLOS ONE

Dear Dr. Tollefson, 

I'm pleased to inform you that your manuscript has been deemed suitable for publication in PLOS ONE. Congratulations! Your manuscript is now being handed over to our production team.

Kind regards, 

on behalf of

Dr. Miquel Vall-llosera Camps 

Staff Editor

PLOS ONE